# Investigation of a 2-DOF Active Magnetic Bearing Actuator for Unmanned Underwater Vehicle Thruster Application

**Muhammad Abdul Ahad and Sarvat M. Ahmad ***

Faculty of Mechanical Engineering, Ghulam Ishaq Khan Institute of Engineering Sciences and Technology, Topi 23640, Pakistan; abdulahad@giki.edu.pk

* Correspondence: smahmad@giki.edu.pk

**Abstract:** In this work, a novel application of Active Magnetic Bearing (AMB) is proposed to integrate AMB in the Magnetically Coupled Thruster (MCT) assembly for underwater application. In this study, a 2-Degree-Of-Freedom (DOF) AMB is developed and investigated for the MCT of an Unmanned Underwater Vehicle (UUV). The paper presents the detailed electro-mechanical modeling of the in-house developed AMB system. The intractable problem of rotor suspension and rotation with opposing pairs of electromagnets is considered. A Linear Quadratic Gaussian (LQG) controller is designed and analyzed in the frequency domain for the stabilization of the open-loop unstable AMB for MCT. The performance specifications of the controller, such as reference tracking and disturbance rejection are achieved and evaluated through real-time implementation of the controller. The compensator also performed reasonably well during the dynamic operations, i.e., when the rotor-propeller assembly was spun at 1500 rpm. This rotor speed is needed to generate a thrust of 40–45 N and up to 1 m/s forward velocity, which is necessary to propel the UUV under consideration. By deploying AMB in MCT assembly, it is anticipated that problems associated with the conventional directly coupled thruster operating in harsh underwater environment, such as water ingress into electronics compartment, rusting, lubrication, and vibrations would be eliminated.

**Keywords:** active magnetic bearing; AMB; electro-magnetic actuator; magnetically coupled thruster; modern controller; LQG

## 1. Introduction

Unmanned underwater vehicles are employed for a wide range of industrial, commercial, and scientific applications and are of immense importance. The research of UUVs began in the 1950s [1], to utilize them in a harsh underwater environment where the divers fail to operate, their presence is dangerous and costly [2,3]. UUV usage has significantly increased as they play a vital role in many applications like surveying, inspection, repairing of underwater structures, pipelines, cables, and fish farm monitoring [4–6].

A thruster is a dynamic and critical component of any UUV and the overall performance is primarily defined by the thruster [7]. A thruster not only propels the vehicle but also allows UUV to perform different maneuvers. A thruster consists of different electrical and mechanical parts, i.e., propeller, motor, speed controller electronics, and connecting shaft. A watertight chamber is required for the housing of the electrical components of the assembly. UUV's thruster utilizes a single shaft to connect the motor with the propeller and for construction, watertight assembly dynamic seals are used. These seals have limitations like wear, tear, and friction, which can cause leakages [8]. Leakages in the sealing of thruster represent a huge loss of potentially valuable equipment and a cause of UUVs inefficiency. The shaft sealing of the thruster is critical in maintaining the overall structural integrity and reliability of UUVs. In permanent magnetic coupling (PMC), mechanical power is transmitted by magnetic field lines without any physical connection between the coupling faces. One end of the coupling is connected with the motor (drive end), while the other is connected with the load (driven end), i.e., propeller, as shown in Figure 1. As there is

no physical contact between the drive and the driven end, the electronics in the housing are completely isolated from water. Hence, PMC offers an attractive solution for problems associated with shaft seals.

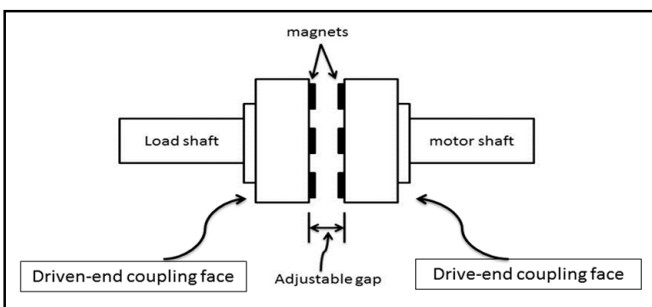

**Figure 1.** Schematic of the PMC mechanism.

The permanent magnetically coupled thruster assembly is shown in Figure 2, the water-tight section slides inside the enclosure and at the junction between the water-tight and exposed section. There is a separating disc, it separates the two faces of the PMC and it seals the drive end of the coupling and the thruster electronics from the driven end. The propeller is mounted on the driven end which is commonly supported by a conventional bearing, as shown in Figure 2. There are many problems associated with the use of the conventional bearing like lubrication, wearing, and rusting as it is exposed to an uncertain underwater environment [9].

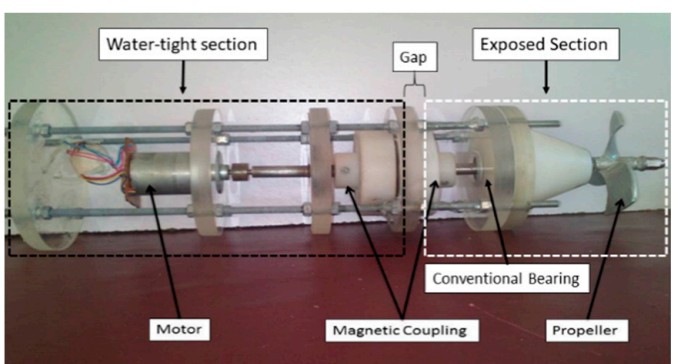

**Figure 2.** Permanent magnetically coupled thruster assembly.

Active magnetic bearings (AMB) are preferred substitute to conventional bearings in a range of applications as the operating environment becomes more extreme, such as extreme temperatures, contamination-free, and corrosive working fluids [10]. Consequently, AMBs are widely employed in high speed rotating machinery, especially in extreme operating conditions [11]. Hence AMB is proposed as a suitable choice for magnetically coupled thruster, to mitigate some of the problems associated with the conventional bearing. AMBs offer advantages like low friction, offers high rotational speed, there is no requirement for lubricants, absence of temperature rise due to non-contact, and auto-adjustment of the center of mass [12–14]. The AMB is utilized in many industrial applications like turbine compressor of the nuclear reactor, where reliability and safety are very important [15], precise machine tools like high-speed spindles of large and miniature milling machines [16,17], wind power generators [18] and medical devices like artificial heart pumps [12] to name a few. An AMB system is a combination of different electrical, mechanical, and electromagnetic components namely, sensors, amplifier, controller, data acquisition card, rotor, and electromagnets (EM). The AMB utilizes the principles of electromagnetism to produce the magnetic field. This generated magnetic field generates bearing forces that act on the rotor, under these electromagnetic forces the rotor is suspended without any physical

contact [19]. The active magnetic bearing systems are inherently open-loop unstable in nature and a feedback controller is needed for its stable and robust operation. The magnetic force produced as a result of the electric current depends on the electric current passing through the coil and the distance between the EM and the rotor. By applying a robust closed-loop controller, the position of the rotor can be accurately controlled even if the load on the suspended rotor is varied as the EMs strength can be regulated [12].

Polajzer [20] described the modeling procedure, reduced-order modeling, and classical Proportional Integral and Derivative (PID) control of a 2-DOF AMB. The actuator sizing, modeling, and PID control of AMB utilizing the mechanical model are presented by Kimman et al. [19]. Polajzer et al. [21,22] designed and implemented PI/PD decentralized controller by considering only the mechanical plant model and also the radial forces non-linearities are studied. Zhong et al. [23] employed a PD controller for position control utilizing a homopolar EM actuator. Dhayani et al. [24] utilized optimized fuzzy-PID controller for AMB stabilization and the controller performance is compared with conventional controllers. Lee et al. [25] proposed a hybrid type AMB design for robotic arm and milling spindle application. However, PID control strategy is employed for stabilization. Jin et al. [26] designed a controller for AMB based on active disturbance rejection theory and the comparison with PID controller is also presented. The second-order sliding mode controller design and analysis for AMB system is presented and its performance is compared with sliding mode controller [27]. The modern control strategy was implemented namely Linear Quadratic Regulator (LQR) with full states information for stable AMB operation [28]. While on the other hand, Jastrzebski et al. [29,30] implemented LQR and a variant of LQR, i.e., LQ/Loop Transfer Recovery (LTR) by considering only the mechanical model. However, for each utility case, the AMB system should be specially designed, fitted, and adjusted to accomplish the desired task in a certain environment [11]. Hence for the proposed underwater thruster application the AMB will be installed in a canned type assembly [31–33]. As the stator and the rotor will be exposed to water and to address the rusting issue, canned type assembly is specifically used for such applications. Moreover, to the author's knowledge to date, no study has been reported on the utility and application of AMB in UUVs and MCTs with several advantages such as high speed, frictionless operation, and no leakage, hence this is the key motivation for the on-going research work. The targets of this study are mathematical modeling, modern feedback control, i.e., Linear Quadratic Gaussian (LQG), test-rig construction, instrumentation, PC interfacing, functional demonstration and experimentation of 2-DOF radial AMB. The established AMB test-rig has four EMs, each opposite pair of EM carry out the operation in differential mode for desirable control of the suspended rotor through modern feedback control. One side of the rotor is positioned inside the EMs, while the other side is connected with the motor drive. The imperative rotor position, damping, and stiffness characteristics are accomplished by feedback control. The stable functioning rotor is demonstrated by a robust LQG controller under different operational conditions.

The paper is structured in a systematic way that Section 2 describes the mathematical modeling while Section 3 describes the 2-DOF AMB closed-loop control. The experimental setup and the experimentation is described in Sections 4 and 5. Experimental results are presented in Section 6 and the paper is concluded in Section 7.

## 2. Mathematical Modeling

AMB system is a complex mechatronic system involving rotor, drive electronics, EM actuators, and displacement sensors. The EM actuators are configured in a way that these actuators function in a differential mode. Through this configuration the rotor can move inside the EMs, i.e., the vertical motion is achieved by actuation of top and bottom EMs, while motion in the horizontal direction is achieved by the exciting left and right EMs. The rotor levitation and the required position is attained by the formation of four EMs, each exerts an attractive force on the rotor. A similar modeling technique is applied for both vertical and horizontal directions. However, for simplification, only vertical position

will be under consideration for this paper and similar logic is applied on the horizontal direction. Nonetheless, the experimental results for both directions will be presented. The dynamic model for the net electromagnetic force of the EMs shown in Figure 3 is given by Equation (1) [34].

$$F_{net} = k \left[ \left( \frac{i_{top}}{s_o - \Delta y(t)} \right)^2 - \left( \frac{i_{bottom}}{s_o + \Delta y(t)} \right)^2 \right] \tag{1}$$

where $k = AN^2 \mu_o / 8$ and it is the electromagnetic constant, where $A$ is the pole area, $\mu_o$ is the relative permeability, $N$ is the number of wire turns, $s_o$ is the nominal air gap, while $\Delta y(t)$ is the change of rotor position from the equilibrium point, while $i_{top}$ and $i_{bottom}$ are the currents of the top and bottom Ems, respectively. The electromagnetic constant incorporates the permeability of the medium in between the rotor and the stator, however, the permeability of air, aluminum, Teflon, and water is almost the same, i.e., $1.25 \times 10^{-6} \frac{H}{m}$ [35], hence the EM is modeled by assuming one medium, i.e., free space rather than with different mediums. The non-linear relationship between net electromagnetic forces-current and net electromagnetic forces-displacement can be seen from Equation (1), hence linear control theory cannot be implemented on this system without linearization of the force model. As the AMB's are operated in differential mode for a fixed value of biased current, $i_{bias}$ and a variable $\Delta i_c$, i.e., controlling current, Equation (1) can be linearized at a certain operating point.

$$i_{top} = i_{bias} + \Delta i_c(t) \tag{2}$$

$$i_{bottom} = i_{bias} - \Delta i_c(t) \tag{3}$$

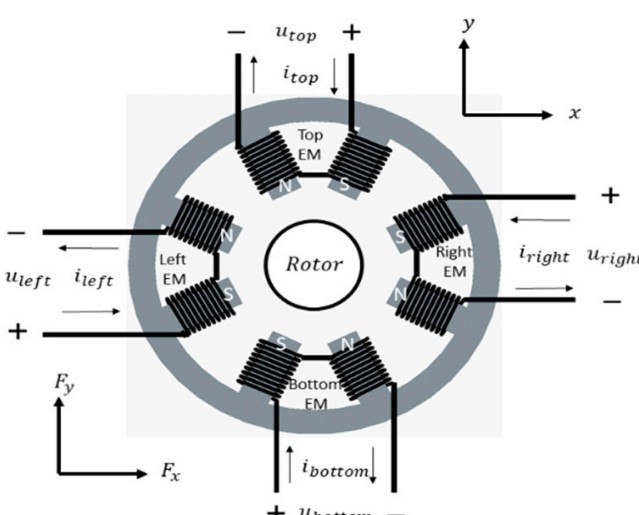

**Figure 3.** Schematic of electromagnets for 2-DOF AMB.

Substituting Equations (2) and (3) in Equation (1), the net EM force becomes

$$F_{net} = k \left[ \left( \frac{i_{bias} + \Delta i_c(t)}{s_o - \Delta y(t)} \right)^2 - \left( \frac{i_{bias} - \Delta i_c(t)}{s_o + \Delta y(t)} \right)^2 \right] \tag{4}$$

When the displacement of the rotor is small, i.e., near operating point or equilibrium point, $\Delta y(t) < s_o$ and dynamic controlling current $\Delta i_c$ is less than the bias current, $\Delta i_c < i_{bias}$. For the above-mentioned conditions for $\Delta y(t)$ and $\Delta i_c$ Equation (4) is approximately linear, after applying Taylor series expansion on Equation (4), the net electromagnetic forces [36] thus becomes;

$$F_{net} = k_i \Delta i_c(t) + k_x \Delta y(t) \tag{5}$$

where $k_i$ the force-current is factor and $k_x$ is the force-displacement factor. Equation (5) is linearized by these factors for certain operating points, i.e., equilibrium rotor position and bias current. The linear force equation is utilized for dynamic modeling and the design of robust controller using modern state space approach employing linear control theory. It is presumed that near the operating point the displacement is very small, by this presumption system can be regulated by employing linear control theory. The rotor with mass $m$ is levitated and tends to displace due to the force generated by the EMs. The force and displacement relationship is given by Newton's second law of motion.

$$F = m\frac{d^2y}{dt^2} \tag{6}$$

The generated bearing forces of EMs are given by Equation (5), hence magnetic bearing is modeled as a linear system by substituting Equation (6) in Equation (5).

$$m\frac{d^2y}{dt^2} = k_i\Delta i_c(t) + k_x\Delta y(t) \tag{7}$$

Applying Laplace transforms on Equation (7) yields:

$$ms^2Y(s) - k_xY(s) = k_iI_c(s) \tag{8}$$

The mechanical plant model for vertical direction is given by,

$$G_{mechanical} = \frac{Y(s)}{I_c(s)} = \frac{k_i}{ms^2 - k_x} \tag{9}$$

The rotor-bearing system's equation of motion is constituted, the next stage is the formation of an electrical model of the AMB system. The pair of the coil is serially connected in each EM, as shown in Figure 3, and by this construction two poles are formed in every EM. The resistance and inductance of all the four EM in this study are the same. The correlation between the current through the coil and the applied voltage is given by,

$$R\Delta i_c + L\frac{d\Delta i_c}{dt} + k_i\frac{d\Delta y}{dt} = \Delta u \tag{10}$$

where $\Delta u$ is net applied voltage on top and bottom EMs, while $R$ and $L$ are the resistance and inductance of the EMs coil, respectively. Applying Laplace transformation on Equation (10)

$$RI_c(s) + LsI_c(s) + k_isY(s) = U(s) \tag{11}$$

Substituting Equation (11) in Equation (8) the electrical plant model for vertical axis is given by,

$$G_{electrical}(s) = \frac{I_c(s)}{U(s)} = \frac{ms^2 - k_x}{mLs^3 + mRs^2 + (k_i^2 - k_xL)s - k_xR} \tag{12}$$

From Equations (9) and (12), the open-loop transfer function in vertical direction is given by Equation (13).

$$G(s) = \frac{Y(s)}{U_x(s)} = \frac{k_i}{mLs^3 + mRs^2 + (k_i^2 - k_xL)s - k_xR} \tag{13}$$

Thus from Equation (13) unstable dynamics of the open-loop AMB system is evident with roots in the right half-plane. Hence, for stabilizing the unstable system, a controller is needed and it is discussed in the subsequent section.

### 3. LQG Control of 2-DOF AMB

In this study, a modern state-space control strategy is employed to design a robust LQG controller for the stable functioning of the AMB system as the AMB system is open-loop unstable. A separate robust LQG compensator is designed for both vertical and horizontal position regulation with a similar design procedure. From the derived open-loop transfer function of the system's characteristic equation, it is evident that the open-loop dynamics of the system is unstable for every value of $K_x$, one pole will always be positive, i.e., in the right half-plane, which in particular represents the unstable dynamic behavior of the AMB system. The closed-loop block diagram representation of the system is shown in Figure 4. Where $k_s$ and $k_a$ are the sensor and amplifier gains, while $Y$ and $Y_{ref}$ are the actual and desired vertical reference positions, respectively.

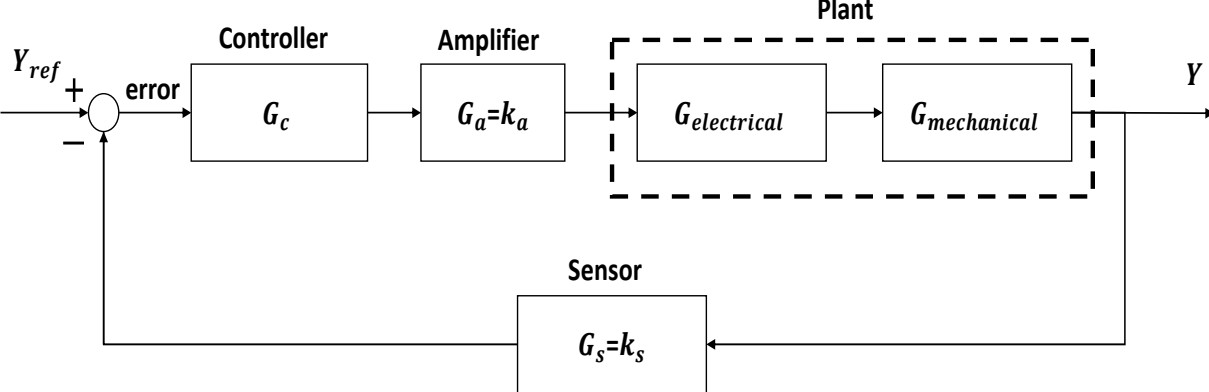

**Figure 4.** Block diagram representation of closed-loop active magnetic bearing system.

For the investigation of the modern state-space control technique, the information of all the states is needed. However, in this case, only the displacement state is obtainable, while the other two states particularly current and velocity are required to be estimated. The state estimator is utilized for the estimation of the unobtainable or missing state information. State estimator in combination with LQR is employed which results in LQG controller. The state-space model of the AMB system conceivably represented as:

$$\dot{x}(t) = Ax(t) + Bu(t) \tag{14}$$

$$y(t) = Cx(t) \tag{15}$$

where $x(t)$ denotes the state vector, $u(t)$ denotes the control vector and $y(t)$ represents the output vector. While $A$ is the state matrix, $B$ is the control matrix and $C$ is the output matrix. The state feedback control law is given by [37]

$$u(t) = -K_{LQR}x(t) \tag{16}$$

where *LQR* controller gain is represented by $K_{LQR}$, which is acquired by the quadratic performance objective function $J$ minimization, with state weighting matrix $Q$ and input weighting matrix $R$. These weighting matrices are changed during an iterative procedure to acquire the required performance characteristics of the controller. The linear system with the plant noises $w$ and measurement noises $v$ is given by [37]

$$\dot{x}(t) = Ax(t) + Bu(t) + w \tag{17}$$

$$y(t) = Cx(t) + v \tag{18}$$

The estimator design yields [33]

$$\dot{\hat{x}}(t) = A\hat{x}(t) + Bu(t) + L_{est}(y(t) - \hat{y}(t)) \tag{19}$$

$$\hat{y}(t) = C\hat{x}(t) \tag{20}$$

the estimator gain, $L_{est}$ obtained by minimizing the cost function with process and sensor noise covariance matrices $Q_w$ and $Q_v$, respectively. While $\hat{x}(t)$ is the estimated state vector and $\hat{y}(t)$ is the estimated output vector, respectively. The LQG compensator, i.e., LQR in combination with the state estimator is shown in Figure 5.

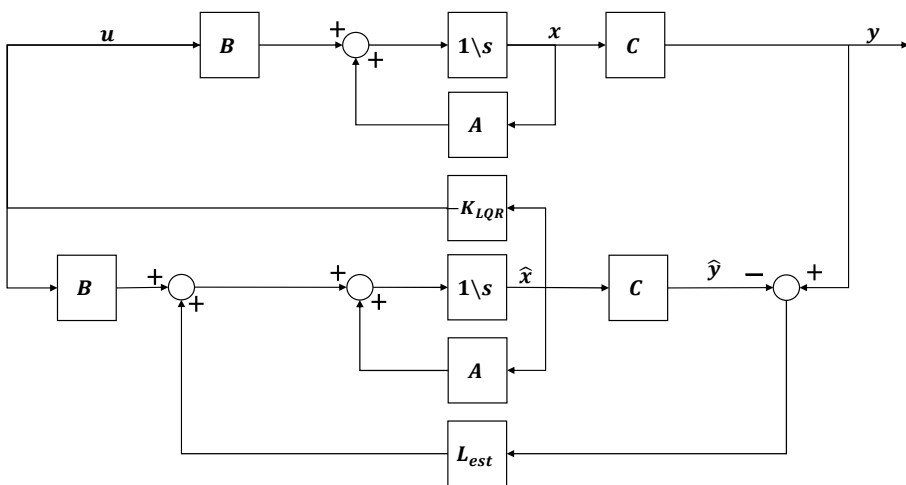

**Figure 5.** Block diagram representation of the state feedback control system with state estimator.

*Analysis*

From the characteristic equation of the AMB plant model, it is observed that the plant is unstable with a pole at 91.9 rad/s in the *y*-plane. In this study frequency response method is employed for the analysis and design of the controller. From Figure 6, it can be seen that at the corner frequency, the gain is less than 0 dB with a phase delay of fewer than $-180$ degrees thus the AMB system is unstable. The compensator is designed for the stable functioning of the AMB system with desired performance specifications the Loop transfer function $L(s)$ is utilized.

$$L(s) = G_C(s) \, G_a(s) \, G_{electrical}(s) G_{mechanical}(s) \, G_s(s) \tag{21}$$

$$G_a(s) = k_a \tag{22}$$

$$G_s(s) = k_s \tag{23}$$

where $G_C(s)$, $G_a(s)$ and $G_s(s)$ are the controller, amplifier, and sensor transfer functions, while $k_s$ and $k_a$ are the sensor and amplifier gains.

For the stable functioning of the closed-loop AMB system, a positive phase margin near the gain crossover frequency is desired. The compensators are designed by employing the frequency response analysis method utilizing an iterative approach. The loop is shaped in a way that for minimal steady-state error and improved settling time, high loop gain is desired at low frequency. However, for robust and stable functioning, positive phase is desired at the crossover frequency. The frequency response for the vertical plane is shown in Figure 7. The phase margin is 17.3 degrees and the gain margin is 8.13 dB. The stable closed-loop system's poles and zeros are illustrated by utilizing complex argand diagram (Figure 8). While the AMB system parameters are listed in Table 1.

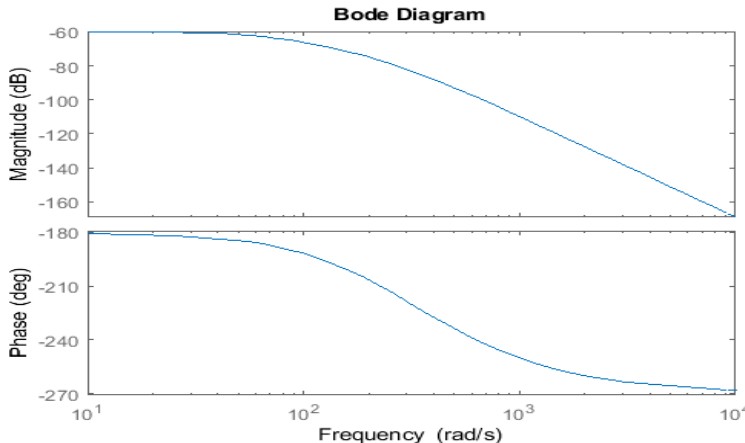

**Figure 6.** The frequency response of AMB plant in *y*-plane.

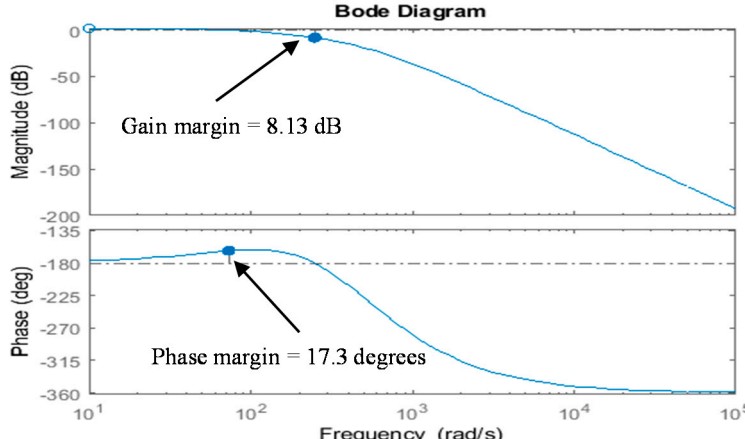

**Figure 7.** Frequency response of the system $L(s)$ in vertical plane *y*.

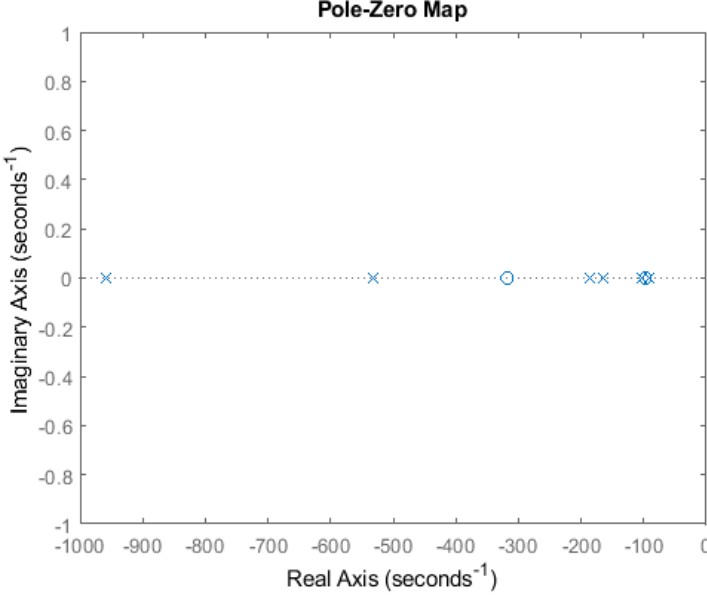

**Figure 8.** Argand diagram of the closed-loop AMB system, X and O are system poles and zeros, respectively.

**Table 1.** AMB system parameters.

| Sr.# | Parameter | Values | Determination |
|------|-----------|--------|---------------|
| 1. | Inductance | 0.014 H | Measured |
| 2. | Resistance | 5 ohm | Measured |
| 3. | Amplifier Gain | 3.41 | Measured |
| 4. | Mass | 343 g | Measured |
| 5. | Nominal Air gap | 1.5 mm | Measured |
| 6. | $i_{bias}$ | 0.98 A | Estimated |
| 7. | $K_x$ | 3252 N/m | Calculated |
| 8. | $K_i$ | 4.928 N/A | Calculated |
| 9. | $K_{LQR}$ | [2973 × 105, 30.6 × 105, 1.3 × 105] | Estimated |

## 4. Experimental Setup

The AMB and the test rig is designed and constructed in house, the configuration of demonstrated AMB rotor is similar to that of a propeller shaft (driven end) of MCT. In this study, a total of four EMs are used which together form a total of eight poles. The rotor is fabricated with ferromagnetic material, hence attractive forces are applied to the rotor when placed inside the EM actuator. The force applied by a EM is $F = B^2 A/2\mu_o$, where $B$ is the magnetic flux density, $A$ is the EM pole area and $\mu_o$ is the relative permeability [38]. For the vertical position regulation top and bottom, EMs generate attractive force, while left and right EMs generate attractive force to regulate horizontal position. In this study heteropolar configuration of the poles is employed, i.e., NS-SN-NS-SN arrangement. The voltages are applied to each EM electronics independently to achieve the desired pole arrangement. The EMs are designed with the requirement to suspend a rotor of 343 g with a clearance of 1.5 mm on each side, i.e., left, right, top, and bottom, respectively. The schematic of the experimental setup is shown in Figure 9.

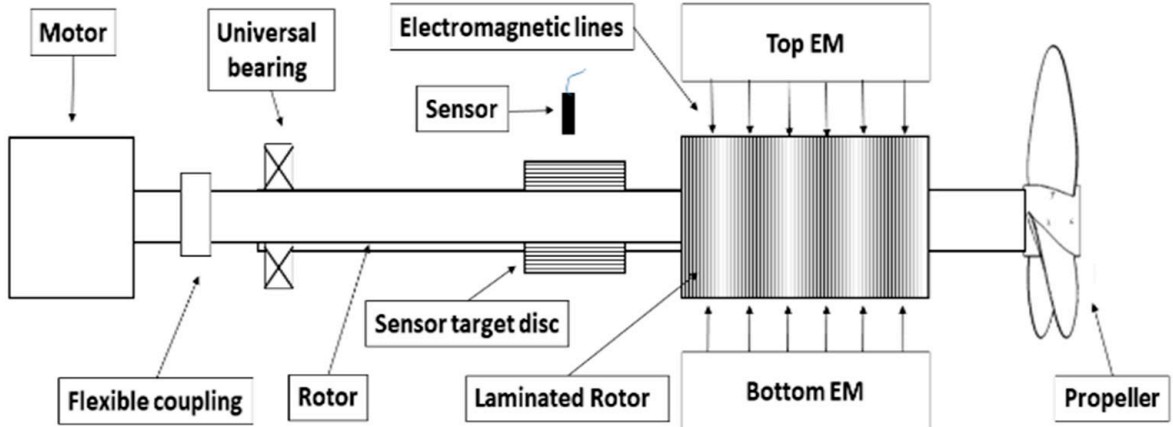

**Figure 9.** 2-DOF AMB experimental setup schematic.

One side of the rotor is placed inside the EMs while the other side of the rotor is connected with the drive motor through a flexible connection, i.e., flexible coupling and universal radial bearing, allowing unconstrained movement in both horizontal and vertical directions and rotation about the z-axis, which represents the pivoted free configuration and through this arrangement 2-DOF is achieved. The side located inside the EMs stator is laminated to keep the eddy-current losses minimum. To prevent the rotor from rusting as it will be exposed to water a thin Teflon sleeve is placed on top, which will prevent direct contact of water with the rotor. The two linear non-contact displacement sensors are utilized to precisely measure the position of the rotor in the vertical and horizontal axis at the free end and are mounted on the dedicated target disc. As the sensors are of inductive type, the ferromagnetic material is used to fabricate the target disc for precise

position measurement. A current amplifier IBT2 with a maximum current rating of 16 A is used and each EM is regulated by an independent amplifier. The EMs are actuated by the pulse width modulated (PWM) signals, the sensors data, and the command signals are received and transmitted by the NI PCIe-6321 DAQ card, the card have 32-bit counters, analog to digital conversion resolution of 16-bit, maximum sample rate of 250 K samples per second, and time resolution of 10 ns. Through the data acquisition card, the commands are transmitted to the actuators and the data from the sensors are received directly in Simulink real-time operating environment. The experimental setup is shown in Figure 10.

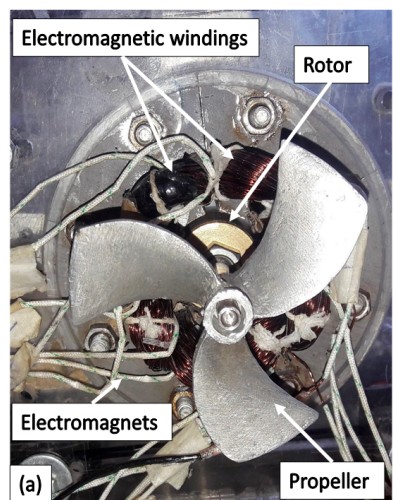 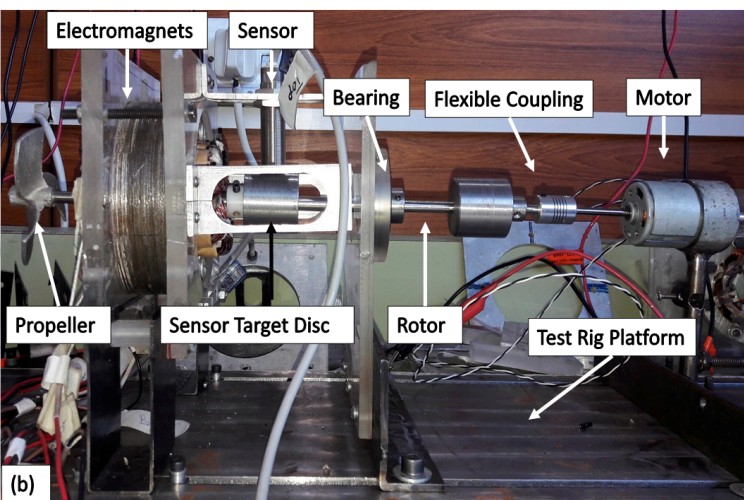

**Figure 10.** Experimental setup of AMB system, (**a**) Front view and (**b**) Side view.

## 5. Experimentation

The designed LQG controller was evaluated for three main performance parameters of the AMB system, i.e., (i) disturbance rejection and reference tracking, (ii) dynamic behavior, i.e., rotation about the z-axis, and (iii) control efforts. The rotor is levitated from its initial position by ramping up to the center position in the bearing. For the evaluation of the disturbance rejection and reference tracking characteristics, external disturbances are applied to the rotor. The dynamic behavior of the AMB is also studied by rotation of the rotor about its lateral axis at different rotational speeds. Real-time data are recorded during the experimentation and analyzed. To achieve desired performance specifications LQG gains were further adjusted during the experimentations.

## 6. Experimental Results

The studied 2-DOF AMB system's experimental results are presented in this section. The performance of the controller is analyzed by the conducted experiments. The three tests were performed, i.e., (i) disturbance rejection and reference tracking; (ii) dynamic behavior, i.e., rotation about the z-axis; and (iii) control efforts. These tests were performed when the levitated rotor is stable at the bearing geometric center, i.e., equilibrium position.

### 6.1. Reference Tracking and Disturbance Rejection

The open-loop unstable rotor-propeller system is stabilized by utilizing LQG controller in real-time. The reference command of 1.5 mm is applied to the rotor, i.e., the geometric center, the subsequent response of the AMB system is shown in Figure 11. From Figure 11, it is evident that, the rotor is not only able to track the commanded reference position but also there is no steady-state error in both axes.

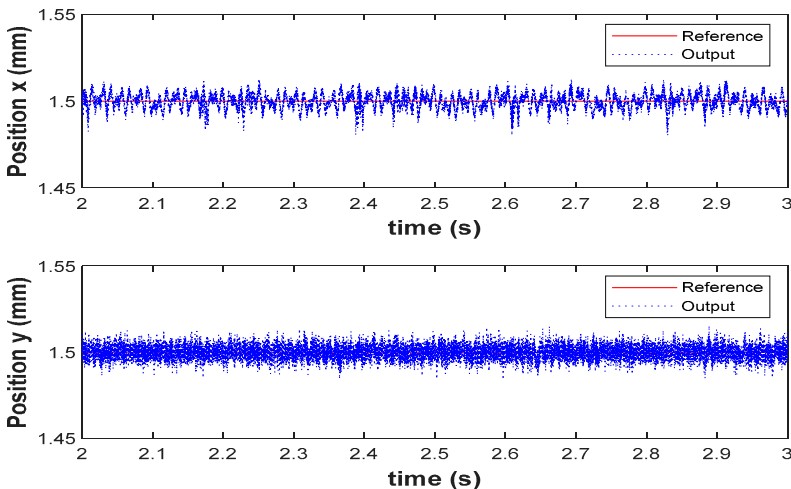

**Figure 11.** Reference command tracking of AMB in both horizontal and vertical planes.

After the tests of reference tracking are performed and the desired satisfactory performance of the controllers are achieved, additionally the disturbance rejection characteristics are assessed. When the rotor was stable and levitated at its geometric center, i.e., reference position, the disturbance rejection capability was tested. The rotor is disturbed by external impulse forces at different time intervals and the response of the rotor position in both vertical and horizontal axes are obtained and are shown in Figure 12.

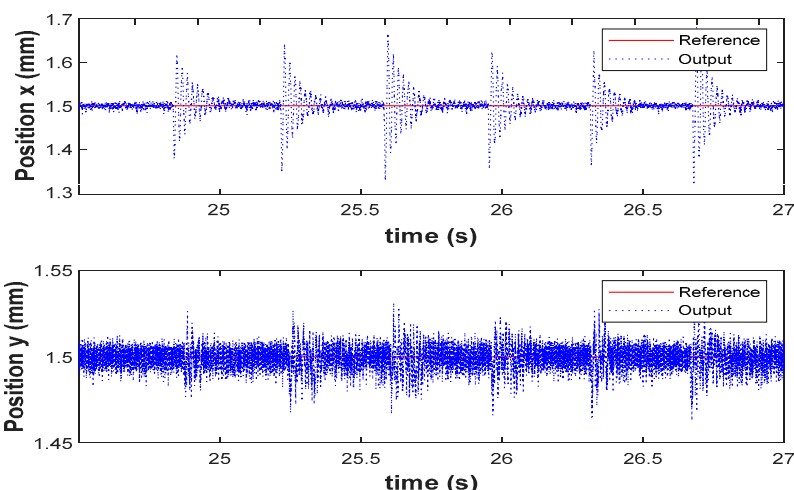

**Figure 12.** Disturbance rejection capability of 2-DOF AMB, when external disturbances are applied.

From the above Figure 12, it can be seen that when the external force is applied to the rotor it is displaced from the reference position but the controllers reject the disturbances and fairly quickly returns the rotor to its equilibrium position.

When the disturbances are applied on the rotor the maximum displacement from its equilibrium position is $\pm 0.12$ and $\pm 0.04$ mm in horizontal and vertical planes, respectively. Nonetheless the system rejects disturbances within 0.15 s and retains its reference position. The designed controllers thus exhibit robust disturbance rejection capabilities of the AMB system, without causing any instability.

### 6.2. Control Efforts

Control current, $i_c$, non-saturation is an important aspect for the design of feedback control algorithm and also for the reliable functioning of the EM actuators. For the compensator design and the linearization of the AMB system dynamics, the basic assumption is $i_c < i_{bias}$, thus enforce a limitation on $i_c$. During the steady-state functioning of the

system, these assumptions must be fulfilled. The control efforts of the AMB system, during reference command tracking, disturbance attenuation, and when subjected to rotation are considered. The control efforts are presented when the rotor is subjected to external disturbances, i.e., impulse forces and the response of the system in terms of position and control current is shown in Figure 13. Figure 13a,b show the position and the respective control current in the horizontal axis. While the vertical axis position and the respective control current is shown in Figure 13c,d. It is evident from the below figure that the bias and the control currents conditions are satisfied for both axes. The control current is responding robustly when the disturbances are applied to the rotor.

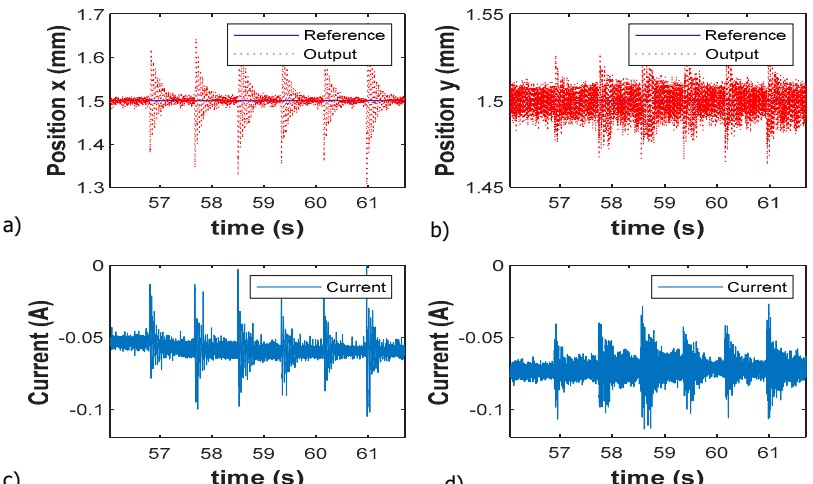

**Figure 13.** The AMB control current response in both *x* and *y* planes, when external disturbances are applied.

The control efforts of the AMB system during reference command tracking at its equilibrium position for both horizontal and vertical planes along with their corresponding control currents are shown in Figure 14, while Figure 15 shows the dynamics response of the system when rotated about its lateral axis.

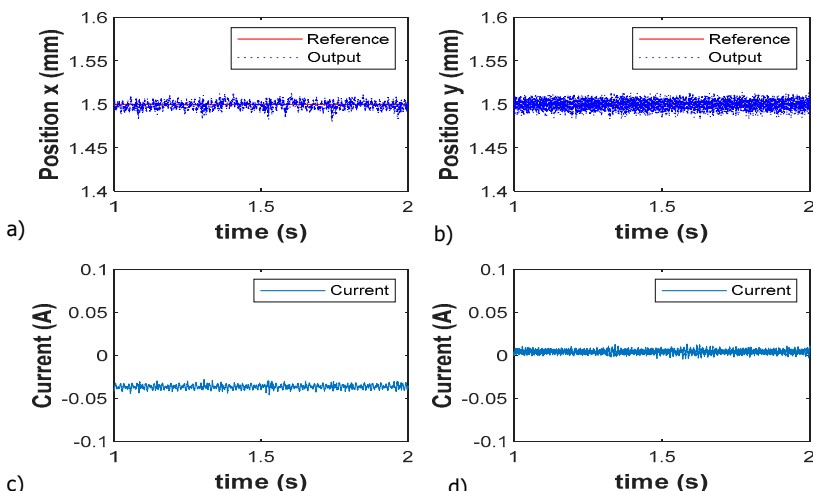

**Figure 14.** The response of control current during reference command tracking of AMB system in both *x* and *y* planes.

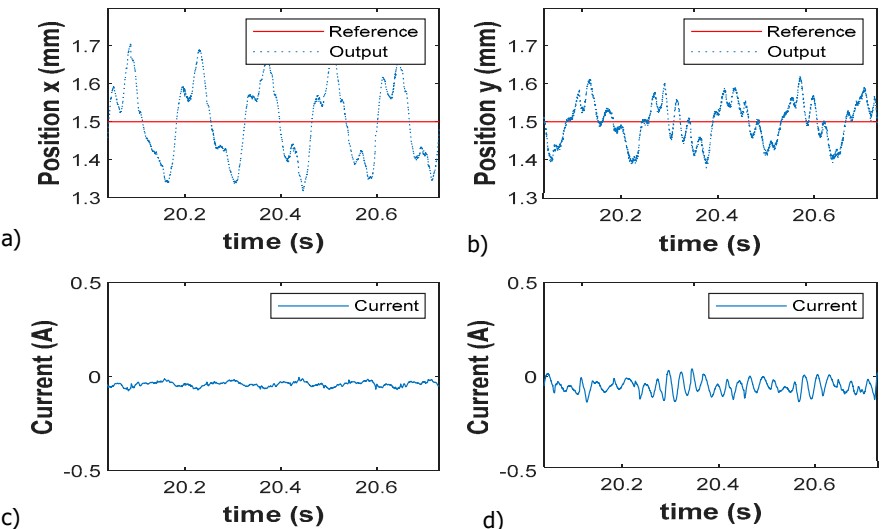

**Figure 15.** The response of control current when 2-DOF AMB is spun at 500 rpm.

It is evident from the above cases, i.e., reference tracking, disturbance rejection, and dynamic response that the control current in both axes is well within the limits of $i_{bias} \pm 0.98$ A. Hence the condition for control current i.e., $i_c < i_{bias}$ is fulfilled.

### 6.3. Dynamic Behaviour

The dynamic behavior of the AMB system is evaluated, while the rotor which is equipped with a propeller is levitated at the equilibrium position and is rotated at different rpm about its lateral z-axis. The dynamic responses of the system at 500 rpm and at 1500 rpm are shown in Figures 16 and 17, respectively. The translation in both *x* and *y* planes can be seen in both the figures, however it is noticeable that the AMB actuator is maintaining the rotor assembly's specified equilibrium position while spinning.

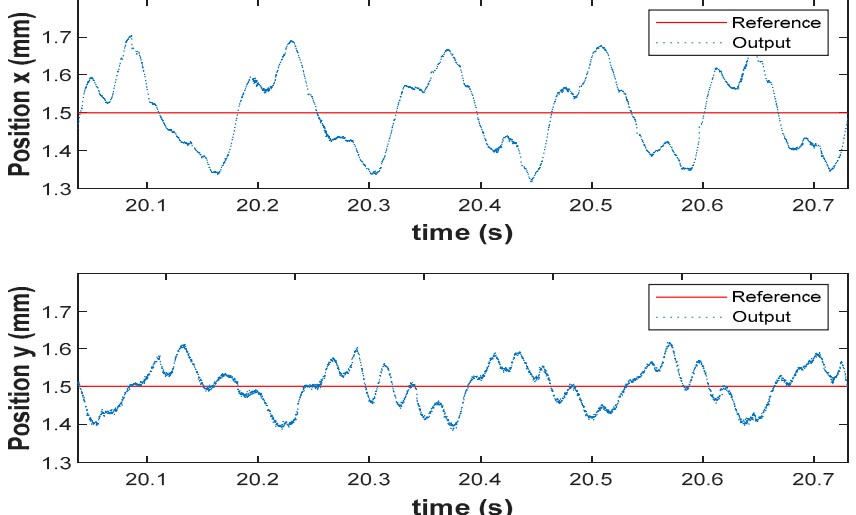

**Figure 16.** The response of AMB at 500 rpm in both *x* and *y* planes.

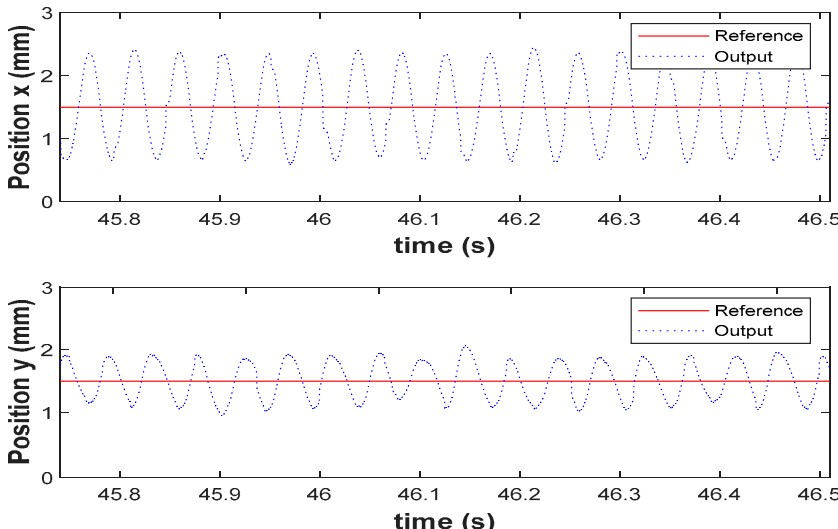

**Figure 17.** The response of AMB at 1500 rpm in both *x* and *y* planes.

The dynamic behavior is also illustrated by the orbit plots. The orbit plot is constituted to have a better understanding and visualization of the rotor's rotational motion and the performance of the controllers during dynamic operation.

The displacement in both vertical and horizontal axes can be seen from the above Figures 18 and 19, the safe vibration boundary of the rotor, i.e., ±1.5 mm in both *x* and *y* planes is represented by circle in these figures. The rotor tries to conserve its reference position with minimal offset while rotating at 500 and 1500 rpm. Though the rotor diverges from the desired position of 1.5 mm still it is in the permissible limit. The rotor drifts about ±0.2 mm from the center in x-plane while in y-plane it drifts around ±0.12 mm, when rotated at 500 rpm. Similarly, when the rotor is rotated at 1500 rpm the drift from the equilibrium position is ±0.9 mm and ±0.55 mm in horizontal and vertical planes, respectively. Nevertheless, the Root Mean Square (RMS) values of control current at different rpm is well within the allowable limits $i_{bias}$ i.e., ±0.98 A.

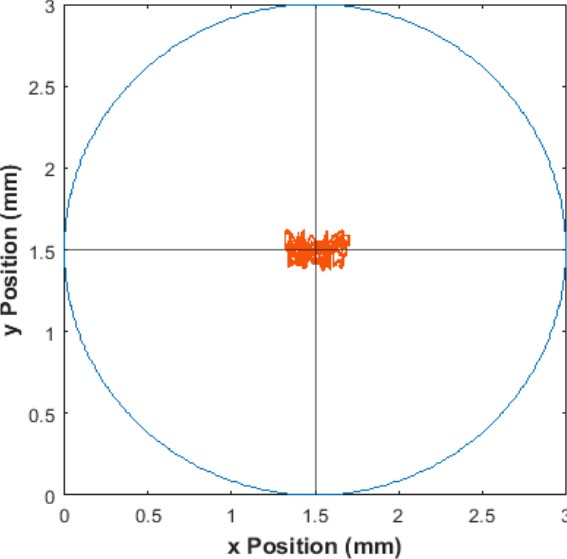

**Figure 18.** Orbit plot of rotor when spun at 500 rpm, circle represents the allowable limit of movement for the rotor.

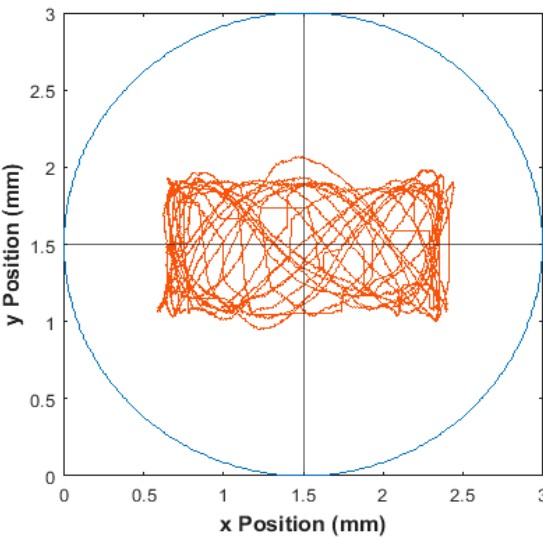

**Figure 19.** Orbit plot of rotor when spun at 1500 rpm, circle represents the allowable limit of movement for the rotor.

The designed LQG controllers are performing considerably well during the dynamic functioning by keeping the rotor within the allowable range of $\pm 1.5$ mm in both vertical and horizontal planes. The MCT is designed to generate a thrust of 40–45 N at 1500 rpm, which is necessary to propel the remotely operated UUV under consideration at around 1 m/s forward velocity.

## 7. Conclusions

In this study, a 2-DOF AMB is developed and investigated for its potential novel application for the MCT of UUV. AMB offers frictionless and lubricant-free operation, hence these can be utilized as an alternative to the conventional bearings used in MCT assembly while addressing pressing problems like water leakage into electronics compartment and attenuation of vibrations. The mathematical modeling of the AMB including the mechanical and electrical along with EM drive electronics is presented in this paper. Modern LQG controllers are employed for the stabilization of the open-loop unstable AMB system. The controllers are designed and also the pre-established performance characteristics are evaluated by frequency response analysis. Through an iterative procedure of frequency response analysis and tuning of the key parameters in real-time resulting in a tuned robust controller satisfying the anticipated performance specifications. The resulting two axes controllers were able to track in real-time the given commands and rejected the applied external disturbances while ensuring that the rotor remained levitated. Finally, the levitated rotor was subjected to a dynamic test by spinning the rotor at 1500 rpm with an acceptable orbit plot. This rotor speed which generates a thrust of 40–45 N, is necessary to propel the UUV under consideration at 1 m/s forward velocity. The proof-of-concept AMB supported thruster rotor demonstration test-rig results are encouraging. Next, the work will be focused on underwater testing for a more realistic evaluation of the AMB supported rotor assembly.

**Author Contributions:** Conceptualization, M.A.A. and S.M.A.; methodology, M.A.A.; software, M.A.A.; validation M.A.A. and S.M.A.; formal analysis, M.A.A.; investigation, M.A.A.; resources, S.M.A.; data curation, M.A.A.; writing—original draft preparation, M.A.A.; writing—review and editing, S.M.A.; visualization, M.A.A.; supervision, S.M.A.; project administration, S.M.A. All authors have read and agreed to the published version of the manuscript.

**Funding:** This research received no external funding.

**Institutional Review Board Statement:** Not applicable.

**Informed Consent Statement:** Not applicable.

**Data Availability Statement:** The data presented in this study is provided in this paper, however any additional data can be provided upon request from the corresponding author.

**Conflicts of Interest:** The authors declare no conflict of interest.

**Abbreviations**

| | |
|---|---|
| AMB | Active Magnetic Bearing |
| UUV | Unmanned Underwater Vehicle |
| MCT | Magnetically Coupled Thruster |
| PMC | Permanent Magnetic Coupling |
| EM | Electro-magnet |
| LQG | Linear Quadratic Gaussian |
| LQR | Linear Quadratic Regulator |
| $F_{net}$ | Net electro-magnetic force |
| $\Delta y$ | Change in rotor position w.r.t $y$ |
| $k_x$ | Force-displacement factor |
| $k_i$ | Force-current factor |
| $L$ | Inductance |
| $R$ | Resistance |
| $i$ | Current |
| $i_c$ | Control current |
| $i_{bias}$ | Bias current |
| $m$ | Mass |
| $s_o$ | Nominal air gap |
| $k_a$ | Amplifier gain |
| $k_s$ | Sensor gain |
| $K_{LQR}$ | LQR gain |
| $L_{est}$ | Estimator gain |

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
