# Peer review of "Investigation of a 2-DOF Active Magnetic Bearing Actuator for Unmanned Underwater Vehicle Thruster Application"

_actuators, doi:10.3390/act10040079_

Round 1
Reviewer 1 Report
This paper describes a new active magnetic bearing. My comments are that the English is poor in places and needs to be looked at, there are a lot of abbreviations, which can cause confusion and the figure captions do not contain enough information.Comments and corrections are below:
- overall the paper needs to be checked for English, it is not good enough in places, which makes it harder to understand the work. Also thing like full stop are missing e.g. line 19 next to the By
- there are a lot of abbreviations within the paper, especially the introduction, this can cause some confusion.
- not sure about abbreviations in the keywords such as "AMB in UUV" could mean anything
- Line 27, don't start the introduction with an abbreviation
- All the figure captions do not contain enough information, they need to include more details of what is seen in the figure, for example Figure 8, what do the X and O represent in the figure these should be in the caption or figure 7 what does the dot on the figures mean. The figure and caption should be stand alone and at the moment they aren't
- Line 66 why is AMB in brackets?
- Line 88 kimman should be a capital K
- Some of the equations such as eqn 8, 10, 12 don't have all the terms defined make sure that all the terms are defined.
- Figure 4 and 5 are not the best quality figures
- equation 22 and 23 have capital K, while the text have little k, are they the same variable or not, please be consistent in the labelling
- Table 1 what are the errors on the measured values, please add them in
- Line 329, in the above figure, need to add a number in, as there are 2 figures above
Author Response
Please refer to the attached letter.

Reviewer 2 Report
Introduction part contains really good state of art in specific research area of UUVs. It focuses mainly on the critical components of thrusters. Thruster consists of different electrical and mechanical parts i.e. propeller, motor, controller electronics, connecting shaft. Active magnetic bearings (AMBs) are a key element of manuscript investigations, also mathematical modelling and feedback control with application of Linear Quadratic Gaussian method. In chapter 2, was created mathematical model of mechanical and electrical part of mechanism. Final form of transfer function was derived from electrical and mechanical plant models. Based on position of roots of characteristic equation in right half plane, its clear this dynamical system is unstable.In next chapter authors designed robust LQR controller in state-space.
In the chapter of experimental setup the authors claim this: for the vertical position regulation top and botton, EMs generate attractive force, while left and right EMs generate attractive force to regulate horizontal position. Based on what do the authors know the direction of the vectors generated by the magnetic field forces? I recommend adding an explanation, mathematical formulation or numerical simulation of the generated electromagnetic field.
I recommend the authors to add the technical specifications of the NI PCIe-6321 DAQ card.
Figures 18 and 19 show orbital graphs showing dynamic behavior. I recommend adding geometric areas of safe vibration boundaries to the graphs, and these graphs can be divided into time sections and color-coded for better analysis of the current state.
I recommend the authors to add references also not older than 3 years.
The paper is suitable for publication in journal Actuators after minor corrections.
Author Response
please refer attached document.
